# Microstructure and Phase Composition of Ti-Al-C Materials Obtained by High Voltage Electrical Discharge/Spark Plasma Sintering

**DOI:** 10.3390/ma17010115

**Published:** 2023-12-25

**Authors:** Rasa Kandrotaitė Janutienė, Olha Syzonenko, Darius Mažeika, Laura Gegeckienė, Ingrida Venytė, Andrii Torpakov

**Affiliations:** 1Department of Production Engineering, Kaunas University of Technology, 44249 Kaunas, Lithuania; darius.mazeika@ktu.lt (D.M.); laura.gegeckiene@ktu.lt (L.G.); ingrida.venyte@ktu.lt (I.V.); 2Institute of Pulse Processes and Technologies, National Academy of Science of Ukraine, 61103 Kharkiv, Ukraine; olgasizonenko43@gmail.com (O.S.); torpakov@gmail.com (A.T.)

**Keywords:** SEM, composites, SPS, high voltage electrical discharge, fullerenes, intermetallic phase, graphene, titanium

## Abstract

Titanium-based composite materials arouse interest in fields like aerospace, transportation, medicine, and other applications. This research project presents the analysis of phase composition of sintered Ti-Al-C composite materials under high voltage electrical discharge. The new technology, described in the previous work of the authors, allows to synthesise the composites containing various intermetallics, carbides, and nanostructures. The samples of Ti-Al-C powder composites were tested by SEM, Raman spectroscopy, and XRD. It was determined that the treatment of the powder by high voltage electrical discharge (HVED) and further sintering at high temperatures using the spark plasma sintering (SPS) method encouraged the formation of the intermetallic reinforcing phases, carbides, and different nanocarbon structures like graphene and fullerenes, as well as pure graphite. Intermetallic phases and nanocarbon structures improved the mechanical and physical properties of the composites. By using the experimental methods mentioned above, the phase composition of Ti-Al-C powder composites obtained at different sintering temperatures was determined. It was revealed that new composite materials produced by HVED and further SPS were rich with carbides, intermetallics, and MAX phases. Therefore, the carbon nanostructures (graphene and graphite) were detected existing in the structure of the produced new Ti-Al-C composite material. All these reinforcing particles improved the microstructure and the mechanical properties of the composites, as was proved in the previous research by the authors and by the different scientific resources. This project is a pilot experimental work, therefore not all peaks of Raman and XRD were detected; they will be analysed in future works.

## 1. Introduction

Titanium and its alloys are rapidly becoming important research objects in various fields such as the automotive and aerospace industries. These materials are lightweight and have excellent properties such as high specific strength, chemical resistance, and biocompatibility. The combination of these attractive properties makes the alloys suitable for structural, chemical, heat resistant, tribological, and biomedical applications [1,2].

Titanium matrix composites (TMCs) are promising metallic materials widely used in various industrial fields, such as the medicine, automotive, and aircraft industries. TMC materials are characterised by exceptional properties, including relatively low density, as well as high levels of specific strength and corrosion resistance [3].

The high modulus of elasticity of titanium matrix composites is the main indicator of their use in airframes, and the high relative strength has been the impetus for their use in the engine industry. Compared with Ti alloys, TMCs normally have better properties such as higher elastic modulus, hardness, strength, elevated-temperature performance, and corrosion resistance due to the addition of reinforcements. The reinforcement particles in TMC system alloys can exist in the form of carbides, nitrides, oxides, borides, and others. For instance, the addition of 5% of TiC can increase the strength of TMC alloy by approximately 20% [4,5].

Titanium-based alloys and composites often contain aluminium due to their excellent properties such as low density, high elastic modulus, high mechanical strength, and high fatigue resistance, which make them widely used in the aerospace, automotive, and transportation sectors. Furthermore, the titanium-based alloys and composites are characterised by a high strength-to-weight ratio [6,7,8].

Carbon, as a component of titanium alloys and composites, is an excellent heat-resistant material. For example, when TMCs containing carbon nanostructures are used at temperatures above 1000 °C, operation duration from 10 to several 1000 h is possible, and sometimes approaches 2200 °C [9]. The presence of carbon in titanium alloys and composites increases the heat resistance of these materials, increases strength, and improves tribological properties. Reinforcing with graphene nanoplates demonstrates excellent properties of TMCs. Graphene nanoplatelets (GNPs) have emerged as “star” reinforcements for achieving excellent mechanical properties in metal matrix composites due to their unique inherent features, including a 1 TPa elastic modulus and 125 GPa fracture strength, good self-lubricity, and low density (1.56–2.08 g/cm^3^) [10].

In Ti-Al-C composites, various processing processes can lead to the formation of the MAX phase, which is a three-layer metal–ceramic material with excellent metal–ceramic properties such as high strength, high hardness, self-lubrication, and good electrical and thermal conductivity [11,12,13]. The ternary carbide Ti_3_AlC_2_ is a typical candidate of the MAX phase and has a multilayered morphology [14].

Titanium aluminides, such as AlTi_3_, Al_3_Ti, or TiAl, can be found in Ti-Al-C composite materials. It was determined that, during heating in molten salts, synthesis of the MAX phase compounds Ti_2_AlC and Ti_3_AlC_2_ proceeded between 900 and 1000 °C with the formation of auxiliary phases TiC and TiAl, which decreased in abundance with rising synthesis temperature [15]. After growing of the synthesis temperature to 1300 °C, Ti_3_AlC_2_ appeared together with the phases Ti_2_AlC and TiC. It was established that the synthesis of Ti_2_AlC allowed the formation of initial TiC_1−x_ on graphite and titanium aluminides [15].

The researchers studied the effect of Ti_3_AlC_2_ content on the physical, mechanical, and arc ablation resistance properties of the produced Cu-Ti_3_AlC_2_ composite and found that Cu-Ti_3_AlC_2_ composite’s properties such as relative density, electrical conductivity, tensile strength, and Brinell hardness were varying depending on the Ti_3_AlC_2_ content in the composites. Samples containing 10, 15, 20, 25, 30, and 35% of Ti_3_AlC_2_ sintered at 800 °C were tested. Since the Ti_3_AlC_2_ phase has intermediate metallic and ceramic properties, increasing its content in the composite increases the mechanical properties and then starts to decrease [16,17].

In another paper [18], researchers analysed the synthesis, deformation, and tribological properties of a new alloy containing Ti_3_AlC_2_-Ti_2_AlC MAX phases. A two-phase composition of MAX-phases was synthesized by SPS in a vacuum environment using Ti, Al, and C primary powders. A layered composite was formed, whose deformation mechanism and tribological properties were studied by SEM, TEM, and Raman spectroscopy. A detailed analysis of the worn surface showed that the Ti_3_AlC_2_-Ti_2_AlC two-MAX-phase composite was self-lubricating.

After reviewing the literature sources, the summarising conclusion can be made that Ti-Al-C and other similar systems are widely used in various fields and are constantly researched by scientists from various countries. These materials have excellent physical and mechanical characteristics. The materials studied in this research are characterised not only by the fact that they contain MAX phases and titanium aluminides (intermetallic compounds) in their structure but also thanks to exposure to a high-voltage electric discharge in kerosene and carbon nanostructures such as graphene, fullerenes, etc., which are less studied. Therefore, the novelty lies in the specific development of technology of powder treatment allowing to produce the new Ti-Al-C composites containing reinforcing particles of carbides, intermetallics, MAX-phases, and carbon nanostructures, with improved physical and mechanical properties. This specific technology involves the treatment of the initial powder (titanium + aluminium) by HVED in kerosene and further consolidation by SPS allowing the synthesis of the reinforcing particles.

This study helps to determine the phase composition and microstructure of Ti-Al-C composite powder materials produced by SPS consolidation of the charge after the HVED processing of titanium and aluminium powders in kerosene.

## 2. Materials and Methods

The samples were prepared from the powder with the following composition: 70–75 mass % of titanium and 15 mass % of aluminium. The prepared mixture of the powder was processed with HVED in liquid kerosene (method is described in detail in [19]). The mass relation of 100 g of hard phase and 1.5 L of kerosene was set to be 1:12. The parameters of HVED were as mentioned here: the stored energy E_1_ = 1 kJ and the integral (total) energy E_sum_ = 25 MJ kg^−1^ that was released during the processing of the disperse system. The heating rate during sintering was 15 deg/°C. The processing schema is presented in Figure 1.

The samples were sintered in vacuum under the pressure of 50–60 MPa in a graphitic die between the graphitic electrodes (Figure 1b). The sintering duration was 5 min for all samples. The temperature and the current used are presented in Table 1. The samples were designated according to the temperature of sintering and the powder composition after HVED.

The consolidation mode of powders of the Ti-Al-C system was selected according to the melting temperature of the most easily melting component, taking T = 0.7 × T_melt_. It was obtained to be 950 °C for samples G13 and G14. For samples 13 and 14, the temperature was increased to 985 °C and 1020 °C to see how this would affect the structure of the material. The prepress pressure in all experiments was ~30 MPa, while the sintering pressure was ~60 MPa.

For the microstructural analysis, the sintered samples (Figure 1c) were mounted on plastic and ground and polished with the different suspensions containing 1 μ, 0.25 μ, and 0.05 μ abrasive particles (supplied by Advanced Abrasives Corporation, Pennsauken, NJ, USA) followed by proper washing. They were explored using a ZEISS EVO MA10 scanning electron microscope (Karl Zeiss Meditec AG, Oberkochen, Germany) equipped with the Bruker XFlash 6/10 EDX detector (Bruker AXS, Karlsruhe, Germany).

The dimensions of cylindrical sintered samples were obtained as Ø8 mm × 5 mm.

An inVia Raman microscope (Renishaw, UK) was used for Raman scattering measurements. The excitation beam was focused on the sample using a 50× objective. Silicon was used for the calibration of Raman setup. Here are the parameters of the laser:-532 nm wavelength.-Laser power at the sample surface—3.5 mW.-Integration time—10 s.-Signal was accumulated 10 times.

The Raman Stokes signal was dispersed with a diffraction grating (2400 grooves/mm) and data were recorded using a Peltier cooled charge-coupled device (CCD) detector (1024 × 256 pixels). This system yielded a spectral resolution of about 1 cm^−1^.

The XRD analysis was carried out on a D8 Advance diffractometer (Bruker AXS, Karlsruhe, Germany). Here are the parameters of the XRD setup:-Tube voltage of 40 kV.-Tube current of 40 mA.-The X-ray beam was filtered with Ni 0.02 mm filter to select the CuKα wavelength (λ = 1.5406 Ǻ).-Diffraction patterns were recorded using a fast-counting detector Bruker LynxEye (Bruker AXS, Karlsruhe, Germany) based on silicon strip technology.-Scanning over the range 2θ = 20–90° at a scanning speed of 6°/min using a coupled two theta/theta scan type.

The XRD data of the samples were analysed, and the quantitative determination of the phases was carried out with the Bruker AXS TOPAS 4.1 [20,21] software program using the Rietveld crystal structure refinement method.

## 3. Results and Discussion

The SEM analysis of the sintered Ti-Al-C powder composites showed that all the samples consisted of a metal matrix and reinforcing particles like carbides, intermetallics, and MAX-phases (Figure 2a–c).

It is visible from the micrographs (Figure 2) that the Ti-Al-C powder composites with a heterogenous microstructure contained a huge number of reinforcing particles in the shape of spheres, plates, and needles. The lowest sintering temperature (950 °C) gave the biggest number of micropores (Figure 2a). With the temperature increased to 985 °C and more, the composite became denser with a much lower number of micropores (Figure 2b–d). In agreement with the scientific article [4], it can be determined that dendritic titanium carbides were formed (elongated plates with different lengths) (Figure 2a,b), and small globular carbide particles were detected, as shown in Figure 1d. The distribution of the elements and phases will be discussed later, presenting SEM mapping microstructures (Figure 3).

The SEM micrographs were taken with a BSE detector. It was determined that there was no significant difference between the obtained images, so it was inappropriate to use the BSE detector further.

Results from the previous research work of the authors [19] showed that strengthening particles were TiC, Ti_3_Al, and Al_4_C_3_ when the composition of the as-received powder after HVED treatment in kerosene was 80% Ti, 15% Al_3_Ti, and 5% Ti_3_AlC_2_. The elemental mapping of the microstructures with different compositions and sintered at different temperatures of SPS allowed to reveal the distribution of every element in the microstructure and to obtain data about possible existing reinforcing particles. The mapping micrographs showed that the needle-like structure contained more aluminium, and the spherical particles were rich with titanium and carbon (Figure 3). Therefore, it can be concluded that the main reinforcing particles can be determined being the carbides of titanium. The shape of these carbides was detected to be globular (Figure 3a,b,g,h) or in the form of small dendrites (Figure 3e,f). The elongated plates, which contained aluminium (blue colour), could be recognised as Al_3_Ti or AlC_2_ particles (Figure 3).

EDS analysis was performed by analysing separate reinforcing particles: spherical and needle-like fractions (Figure 4) and dispersed unknown elements detected at the grain boundaries—possible titanium carbides—located at the grain boundaries [22]. It was determined that the reinforcing particles mainly consisted of titanium; however, peaks of carbon and aluminium were visible, especially when the elongated crystal plates were investigated. These crystal plates could be phases of titanium and aluminium [23].

Some micrographs showed contamination with iron, possibly due to the presence of steel electrodes (Figure 4a).

Raman spectroscopy was performed on the dark carbide particles, dark big areas, and light grains of the matrix. Raman spectroscopy analysis of the results was made by comparing the values of the intensity of the peaks obtained and the ones found in the literature resources. The review is presented in Table 2.

The results of Raman spectroscopy analysing the Ti-Al-C powder composites are presented in Figure 5.

The results of Raman spectroscopy showed that the dark areas of the Ti-Al-C powder composites revealed the peaks corresponding to the carbon nanostructures like fullerenes C_60_ and C_70_, graphene, or crystalline graphite (Figure 5a,b). The strong peak at about 1580 cm^−1^ was detected in all samples, but it was hard to indicate the nature of the carbon nanostructure since some of the literature resources state that this peak corresponds to the graphene [32,34] and others to the crystalline graphite [25,35]. The literature resource [30] proves the existence of graphene, which presents a peak at about 1555 cm^−1^. Graphene is believed to be one of the most promising materials to reinforce the metal matrix composites due to its strengthening of the sintered powder materials [24,26,32,34,37].

The areas with small reinforcing particles showed different characters of the Raman curves (Figure 5c,d). The figures present strong peaks at 1555 cm^−1^ and 1570 cm^−1^, which could be obtained due to the graphene [28,30,31].

The peaks observed in the region between 2300 cm^−1^ and 3000 cm^−1^ referred, probably, to the Ti-C alloy, with the exception of the peak at about 2713 cm^−1^, which was the overtone of the peak at 1355 cm^−1^ (in our case 1346 cm^−1^ and 1351 cm^−1^) [36]. One sample showed the peak at about 632 cm^−1^ (Figure 5c), which could refer to Ti_3_AlC_2_ [23]. Furthermore, it was determined that the MAX phase Ti_2_AlC, which occurred below 900 °C, transformed into MAX phase Ti_3_AlC_2_ when the temperature increased above 900 °C [17]. In this case, the sintering temperature was from 950 °C to 1020 °C, so the existence of Ti_3_AlC_2_ could be possible.

Comparing the obtained results with the data found in scientific resources, it could be summarised that the proposed method of HVED treatment in the kerosene including SPS enabled to obtain the metal matrix composites reinforced with the carbon nanostructures that develop mechanical, electrical, and thermal properties, such as heat and wear resistance [32,34,38].

The detailed phase composition was obtained by X-ray diffraction using Rietveld calculations (Figure 6) [39].

Presented amounts of the phases are shown in weight percentage. It is worth noting that graphene cannot be detected by X-ray diffraction. Graphene is composed of a single layer (monolayer) of carbon atoms tightly bound together in a hexagonal lattice [37]. This is an allotropy of carbon. When graphene layers stack on top of each other, graphite is formed. Separated graphene layers in graphite are retained by Van der Waals forces, which can be broken up during exfoliation. Then, the graphene is obtained. At least two parallel planes of atoms are required for detection by X-ray diffraction. Therefore, graphene was almost undetectable in the X-ray diffractogram. So, where graphite was identified by XRD, it was graphite. The most intense maximum 2TH = 26.40 degrees showed that the graphene layers were already packed into the graphite lattice.

Considering the carbon nanostructures detected in the Ti-Al-C powder composites, it could be proved that different allotropies of carbon were found in the microstructure: graphene, fullerenes, and graphite.

Two types of the intermetallic phases and one type of carbide were detected existing in the microstructure: AlTi_3_, Al_3_Ti, and TiC, respectively. The curves of the diffraction analysis are presented in Figure 6, and the crystal parameters of the phases detected are listed in Table 3.

Samples G13 and G14 showed a lot of graphite formed in the composition of the material. Then, the hardness would decrease as it did not react with titanium. This meant that, when processing the powders, the specific energy was higher than necessary and excess graphite appeared. It was necessary to regulate the amount of energy during processing so that there would be no excess graphite and it could all react during processing, as well as the small amount that remained during sintering. Therefore, the parameters of the technology of HVED + SPS should be improved.

Despite the high content of the graphite, the microhardness measured in the previous work of the authors [19] was obtained from 400 to 600 HV. This meant that the graphite was equally distributed and did not decrease the mechanical properties of the material.

XRD analysis of the samples showed that they were composed of different amounts of intermetallics like Al_3_Ti and AlTi_3_ (Figure 6). These materials have high melting points, high mechanical strength, oxidation resistance, and low densities [40]. The phase composition of the different samples treated under different technological parameters obtained from XRD results are presented in Figure 7. Comparing the contents of Al_3_Ti and AlTi_3_, AlTi_3_ was dominant, which was less stable than the Al_3_Ti.

## 4. Conclusions

This research focused on the dependence of the final phase composition and microstructure on the SPS parameters. This study investigated the Ti-Al-C composite powder materials produced by SPS consolidation of the powder mixture after HVED processing of titanium and aluminium powders in kerosene. An experimental plan was developed that allowed to establish the relationship between experimental parameters and results. It was determined that the temperature of SPS had influence on the microstructure and phase composition of the material. Increasing temperature stimulated the decreasing of the excess carbon and less stable Ti_3_Al, while the content of stable TiC, Ti, and Al_3_Ti slightly rose. Therefore, such findings can be summarised:The SEM analysis showed the microstructure with a big number of reinforcing particles (shape of globular particles and elongated needles distributed randomly in clusters all over the surface as a matrix). Samples had few pores on their surfaces. The lower temperatures made the formation of more pores. The elemental mapping of the Ti-Al-C composite powder materials allowed to determine the distribution of Ti, Al, and C in the microstructure. It was observed that the spherical particles consisted mainly of titanium and carbon since the elongated grains revealed mostly aluminium and less titanium. Here, the line intensity of carbon was obtained to be low. Probably, the phases of TiC and Ti_3_AlC_2_ were detected, followed by the carbides and MAX-phases, respectively.Raman analysis of the composites revealed the existence of carbon nanostructures in the microstructure. The dominant carbon nanostructures were graphene and fullerenes C_60_ and C_70_. These nanostructures increased the strength and tribological properties of the material. Some detected peaks were not determined and will be investigated in further studies.XRD analysis showed that samples contained phases like TiC, AlTi_3_, and Al_3_Ti that strengthened material and increased heat resistance of the composites. The samples showed the formed different amount of graphite depending on the sintering temperature (increasing temperature decreased the excess graphite).Considering the carbon nanostructures detected in the Ti-Al-C powder composites, it could be concluded that different allotropies of carbon were found in the microstructure (graphene, fullerenes, and graphite). The previous investigation showed that the microhardness was sufficient (400–600 HV), meaning that the graphite was distributed equally and did not decrease the mechanical properties of the material. Despite that, the technological parameters of the HVED + SPS procedure should be improved.

Further investigation of Ti-Al-C composites will be performed according to the tasks of the joint project of Lithuania and Ukraine.

## Figures and Tables

**Figure 1 materials-17-00115-f001:**
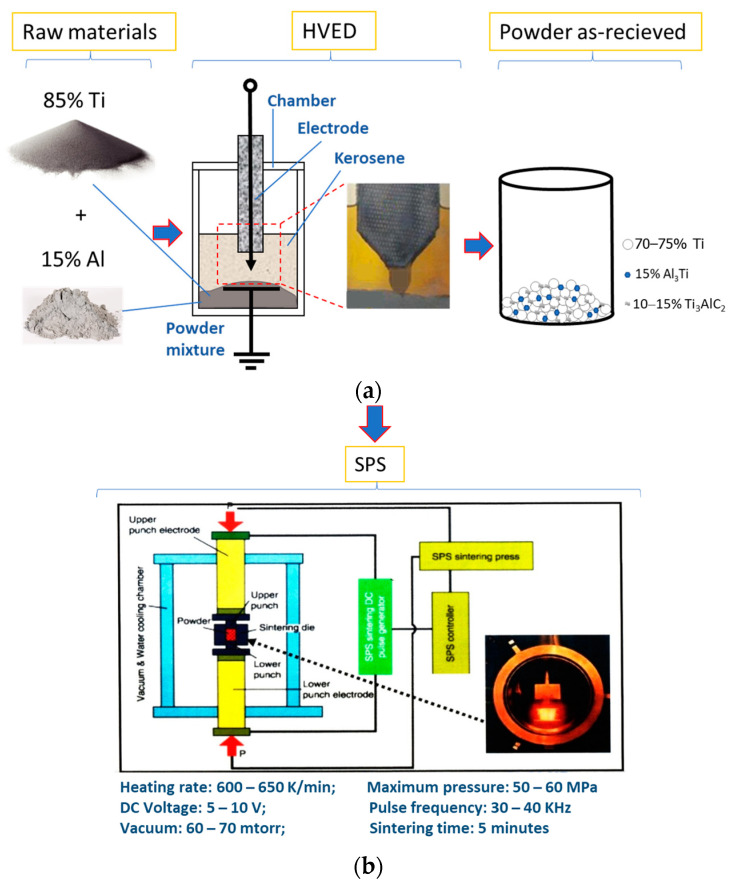
Schema of the preparation of the Ti-Al-C composite powder samples: (**a**) powder treatment by HVED; (**b**) sintering of the samples after HVED; (**c**) samples as received.

**Figure 2 materials-17-00115-f002:**
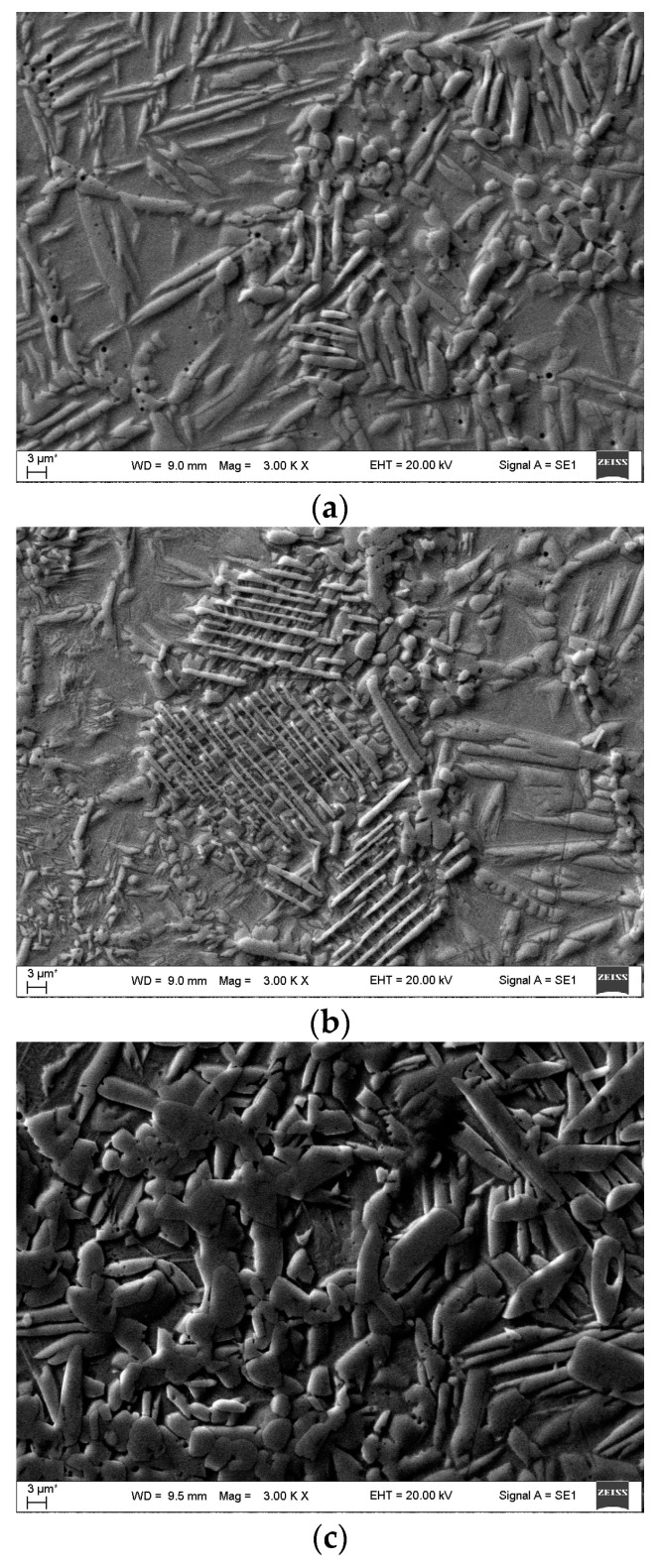
Metal matrix and reinforcing particles (carbides, and intermetallic phases) by SEM analysis: (**a**) sample G13, sintered at 950 °C; (**b**) sample G14, sintered at 950 °C; (**c**) sample 13, sintered at 985 °C; (**d**) sample 14, sintered at 1020 °C.

**Figure 3 materials-17-00115-f003:**
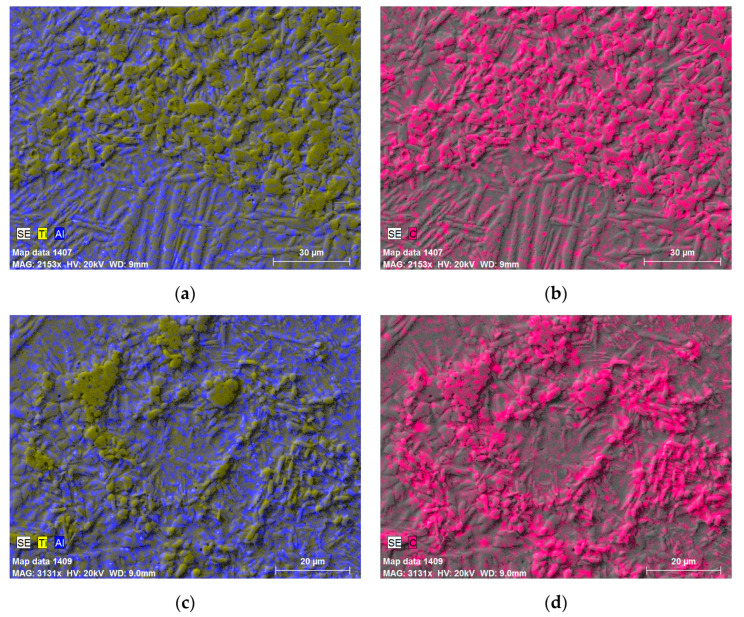
The elemental mapping of the samples: (**a**,**b**) sample G13, sintered at 950 °C; (**c**,**d**) sample G14, sintered at 950 °C; (**e**,**f**) sample 13, sintered at 985 °C; (**g**,**h**) sample 14, sintered 1020 °C.

**Figure 4 materials-17-00115-f004:**
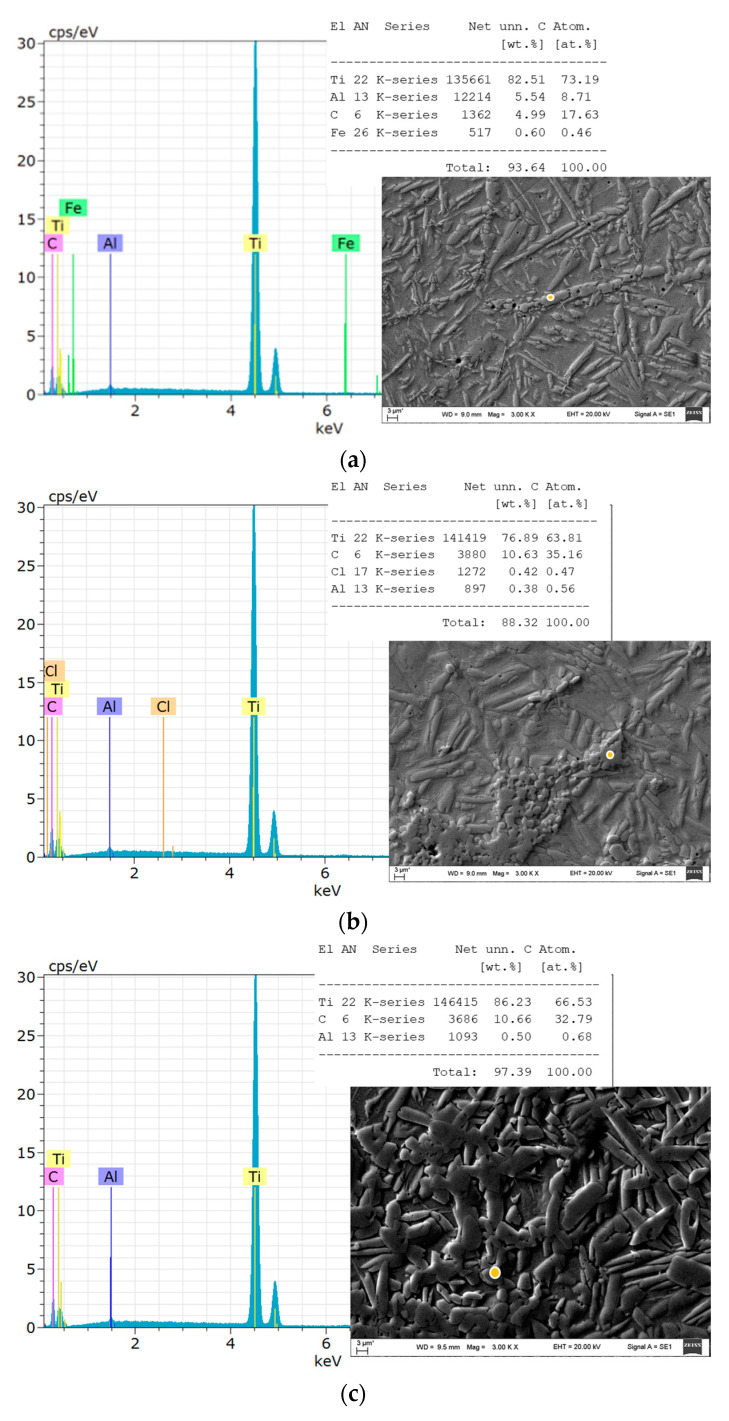
EDS analysis of the samples: (**a**) sample G13, sintered at 950 °C; (**b**) sample G14, sintered 950 °C; (**c**) sample 13, sintered at 985 °C; (**d**) sample 14, sintered at 1020 °C.

**Figure 5 materials-17-00115-f005:**
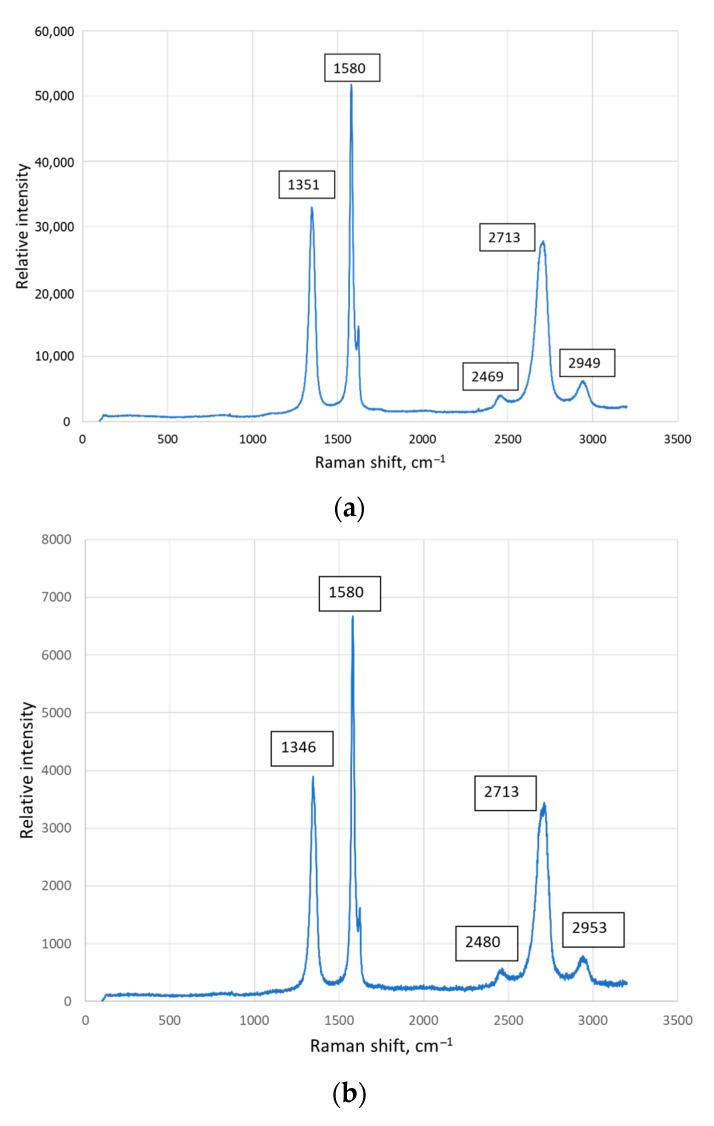
Raman analysis of the samples with a composition: (**a**) 75% Ti, 15% Al3Ti, 10% Ti3AlC2 sintered at 950 °C; (**b**) 70% Ti, 15% Al3Ti, 15% Ti3AlC2 sintered at 950 °C; (**c**) 75% Ti, 15% Al3Ti, 10% Ti3AlC2 sintered at 985 °C; (**d**) 70% Ti, 15% Al3Ti, 15% Ti3AlC2 sintered at 1020 °C.

**Figure 6 materials-17-00115-f006:**
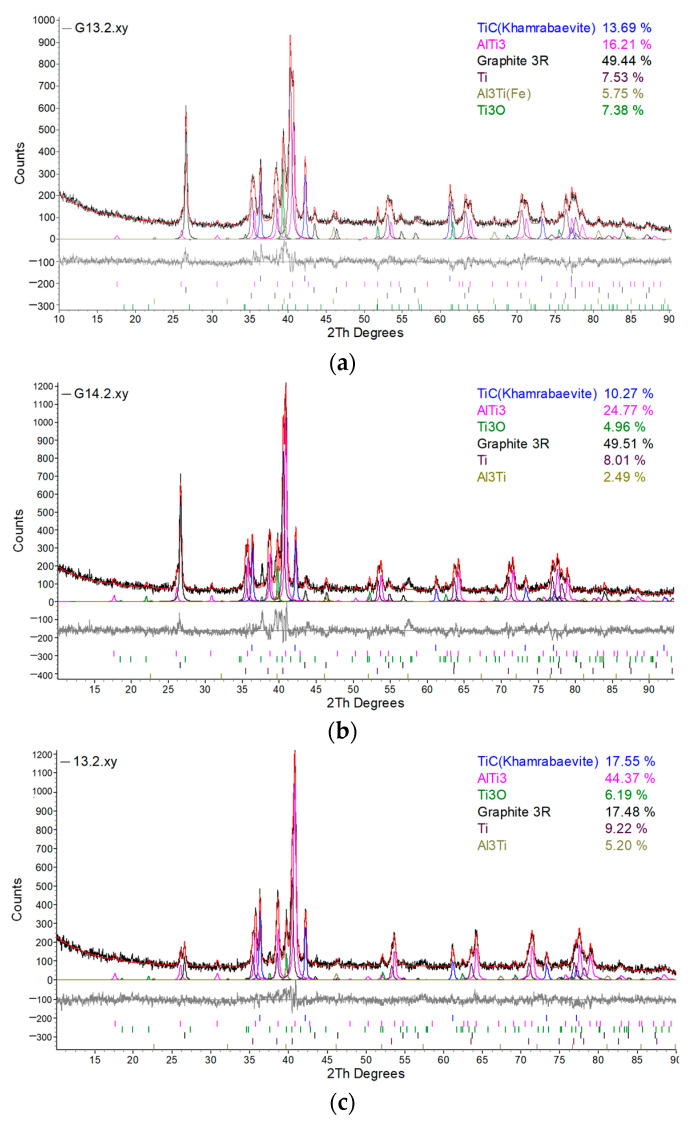
XRD analysis of the samples (red line means the curve taken from the sample): (**a**) sample G13, sintered at 950 °C; (**b**) sample G14, sintered at 950 °C; (**c**) sample 13, sintered at 985 °C; (**d**) sample 14, sintered at 1020 °C.

**Figure 7 materials-17-00115-f007:**
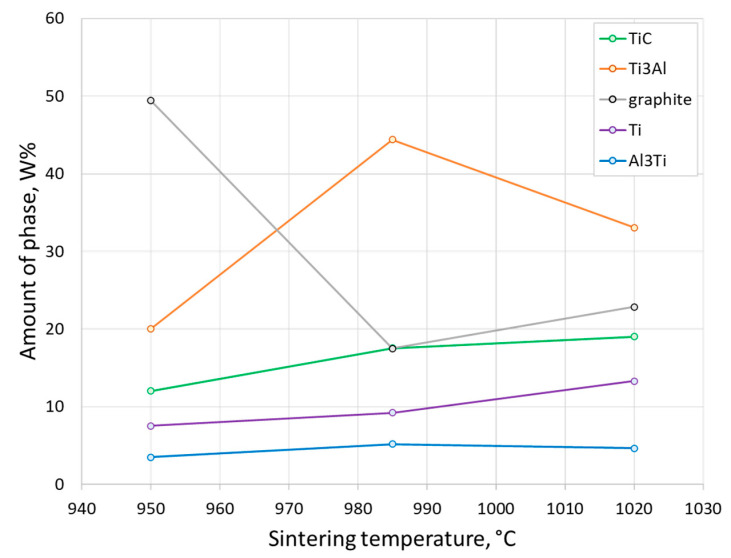
Content of phase composition of Ti-Al-C powder materials depending on the sintering temperature.

**Table 1 materials-17-00115-t001:** Data SPS of the powder samples under HVED.

Designation of Specimens	Parameters of SPS			Composition of the Powder after HVED Treatment
	T, °C	I, A	t, min	
G13	950	840	5	75% Ti, 15% Al_3_Ti, 10% Ti_3_AlC_2_
G14	950	770	5	70% Ti, 15% Al_3_Ti, 15% Ti_3_AlC_2_
13	985	995	5	75% Ti, 15% Al_3_Ti, 10% Ti_3_AlC_2_
14	1020	915	5	70% Ti, 15% Al_3_Ti, 15% Ti_3_AlC_2_

**Table 2 materials-17-00115-t002:** Identification of the peaks observed in Raman spectra.

	Position of the Peak or Group of the Peaks, cm^−1^	Identification of the Peak	References
	125	Ti_3_AlC_2_	[17]
1	148–154	Al-C alloy	[24,25,26]
	150	Ti_2_AlC	[17]
	183, 201	Ti_3_AlC_2_	[17]
2	208–213	Al(Ti)-C alloy	[24,25,26]
3	256–267	Al(Ti)-C alloy	[24,25,26]
	262, 268	Ti_2_AlC	[17]
4	271–273	C_60_	[27]
	270	Ti_3_AlC_2_	[17]
	365	Ti_2_AlC	[17]
5	415–431	Al(Ti)-C alloy	[24,25,26,27]
6	488–497	C_70_	[27,28]
7	608–613	Al(Ti)-C alloy	[24,25,26,29]
	623	Ti_3_AlC_2_	[17]
8	632–633	Ti_3_AlC_2_	[23]
	663	Ti_3_AlC_2_	[17]
9	771–773	C_60_	[27,30]
10	820–840	Al_4_C_3_	[24]
11	1234–1252	C_60_	[27,30]
12	1331	Graphene (D band)	[31]
13	1340	Graphene (D band)	[30]
14	1350	Graphene (D band)	[32,33,34]
15	1355–1360	C_60_	[26]
16	1419–1422	C_60_	[27]
17	1462–1469	C_60_	[27,28]
18	1555	Graphene (G band)	[31]
19	1575–1576	Graphene (G band)	[28,30]
20	1580–1583	Crystalline graphite	[27,35]
21	1580–1600	Graphene (G band)	[32,34]
22	1584–1590	C_70_	[29,33]
23	2700–2720	Overtone of the line at 1355 cm^−1^	[34,36]
24	2930–2950	Ti-C alloy	[36]
25	2320	Ti-C alloy	[36]
26	2690–2700	Graphene (2D band)	[30,32,35]

**Table 3 materials-17-00115-t003:** Quantitative analysis of Rietveld and lattice parameters of samples.

Characteristics	Sample 13					
	Composition 75% Ti, 15% Al3Ti, 10% Ti3AlC2					
	TiC	AlTi3	Ti3O	Graphite	Ti	Al3Ti
Lattice parameters, Å	a = b = c = 4.2807911	a = 5.7986173 c = 4.6547830	a = 5.1486057 c = 9.5700150	a = 3.6350000 alpha(°) = 39.49	a = 2.9259843 c = 4.6713295	a = 3.9302024
Lattice system	cubic	hexagonal	tetragonal	hexagonal	hexagonal	tetragonal
Wt%-Rietveld	17.615	44.490	6.229	17.241	9.261	5.165
Characteristics	Sample 14					
	Composition 75% Ti, 15% Al3Ti, 10% Ti3AlC2					
	TiC	AlTi3	Ti3O	Graphite	Ti	Al3Ti
Lattice parameters, Å	a = b = c = 4.2839790	a = 5.7972892 c = 4.6517540	a = 5.1498631 c = 9.5473012	a = 3.6350000 alpha(°) = 39.49	a = 2.9073658	a = 3.9480215
Lattice system	cubic	hexagonal	tetragonal	hexagonal	hexagonal	tetragonal
Wt%-Rietveld	19.009	33.068	7.150	22.840	13.288	4.644
Characteristics	Sample G13					
	Composition 70% Ti, 15% Al3Ti, 15% Ti3AlC2					
	TiC	AlTi3	Ti3O	Graphite	Ti	Al3Ti
Lattice parameters, Å	a = b = c = 4.2798373	a = 5.8248289 c = 4.6642543	a = 5.2137457 c = 9.6088701	a = 3.6350000 alpha(°) = 39.49	a = 2.9400951 c = 4.6976340	a = 3.9484588
Lattice system	cubic	hexagonal	tetragonal	hexagonal	hexagonal	tetragonal
Wt%-Rietveld	13.689	16.209	7.384	49.441	7.533	5.744
Characteristics	Sample G14					
	Composition 75% Ti, 15% Al3Ti, 10% Ti3AlC2					
	TiC	AlTi3	Ti3O	Graphite	Ti	Al3Ti
Lattice parameters, Å	a = b = c = 4.2835815	a = 5.8025951 c = 4.6481228	a = 5.1486057 c = 9.5700150	a = 3.6350000 alpha(°) = 39.49	a = 2.9253779 c = 4.6750103	a = 3.9300000
Lattice system	cubic	hexagonal	tetragonal	hexagonal	hexagonal	tetragonal
Wt%-Rietveld	10.268	24.766	4.961	49.506	8.012	2.487

## Data Availability

Data are contained within the article.

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
