# Peer review of "Microstructure and Phase Composition of Ti-Al-C Materials Obtained by High Voltage Electrical Discharge/Spark Plasma Sintering"

_materials, 2023, doi:10.3390/ma17010115_

Round 1

Reviewer 1 Report

Comments and Suggestions for Authors

1.     Avoid “I, we” in the manuscript. Please carefully check English grammar and improve the contents.

2.     Half of the literature is not up to date. Please improve this.

3.     On page 4 table 1. Why did authors select these parameters? Why not keep the I and T same for G13 & G14? and/or 13 & 14?

4.     On page 6, the explanation for X-ray results is not clear and sufficient. Especially from line 152 onwards.

5.     On page 7, the scale for EDS map is not the same. All the samples should be compared under the same scale. In addition, it is suggested to keep the color for each element the same (for examples, pink for Al, green for Ti, etc.).

6.     For conclusion, it is suggested to use bullets explaining the findings.

Comments on the Quality of English Language

There are some English grammar mistakes. Please improve the manuscript by native speaker.

Author Response

Dear reviewer,

Please, find the corrections, which are highlighted in yellow. Thank you for your valuable time!

Reviewer 2 Report

Comments and Suggestions for Authors

l 39, 'at 760 °C are 50 % lighter': The lightness has nothing to do with temperature.

l 57, 'Ti2AlC' is doubled

l 71 'Ti3AlC2-Ti2AlC' is not a metal-ceramic composite, the authors named it as 'dual MAX phase composite'

introduction: Please add at the end of the introduction section which kind of material do you want to produce. Should it be a metal matrix with reinforcing particles of carbides and intermetallics? What is then the indended maximum amount (volume) of the reinforcing particles? Or do you want to create a MAX phase composite as in reference [17]? The motivation of the work is therefore not so clear.

l 90: Please give the amounts either in mass or molar percent, not only percent.

table 1: Which heating/cooling rate was used? Please add.

figure 2: Do you have SEM micrographs taken in BSE contrast? It would be more helpful to identify the phases (together with EDS and XRD measurements) as only the ones in SE contrast.

figure 6, XRD results: Are the obtained phase contents in mass or volume percent?. Give also references for the used structure models of the Rietveld analysis.

figure 7: Do not use a spline to connect the points, it gives a wrong impression of the trends.

Comments on the Quality of English Language

There were several places where you wrote  'the MAX phase' without naming any. In such cases it would be better to use the more general term 'MAX phases' instead.

Author Response

(The authors gave the same response as above.)

Round 2

Reviewer 1 Report

Comments and Suggestions for Authors

Some of the issues have been addressed, however, there are still contents needed to be improved. For example, on page 4 line 129-131, the format of the paragraph should be checked and improved.

In addition, the scale of Figure 4 is still small and not the same. There are scales of 4, 9, 10 μm, which means the magnification of the images were not the same. The samples should be compared under the same condition.

Comments on the Quality of English Language

Majority is ok. It is suggested to check the format.

Author Response

Dear reviewer,

Thank you for your valuable remarks trying to do our manuscript better!
